# CAMELS-GB v2: hydrometeorological time series and landscape attributes for 671 catchments in Great Britain

Gemma Coxon<sup>1</sup>, Yanchen Zheng<sup>1,2</sup>, Rafael Barbedo<sup>3</sup>, Hollie Cooper<sup>3</sup>, Felipe Fileni<sup>4</sup>, Hayley J. Fowler<sup>4</sup>, Matt Fry<sup>3</sup>, Amy Green<sup>4</sup>, Tom Gribbin<sup>5</sup>, Helen Harfoot<sup>6,7</sup>, Elizabeth Lewis<sup>8</sup>, Germano Gondim Ribeiro Neto<sup>1</sup>, Xiaobin Oiu<sup>4</sup>, Saskia Salwey<sup>2,9</sup>, Doris E. Wendt<sup>1,10</sup>

- <sup>1</sup> School of Geographical Sciences, University of Bristol, Bristol, UK
- <sup>2</sup> School of Civil, Aerospace and Design Engineering, University of Bristol, Bristol, UK
- <sup>3</sup> UK Centre for Ecology & Hydrology, Maclean Building, Crowmarsh Gifford, Wallingford, United Kingdom
- <sup>4</sup> School of Engineering and Tyndall Centre for Climate Change Research, Newcastle University, Newcastle upon Tyne, United Kingdom
- <sup>5</sup> British Geological Survey, Keyworth, United Kingdom
- <sup>6</sup> Environment Agency, Romsey, Hampshire, United Kingdom
- <sup>7</sup> School of Geography and Environmental Science, University of Southampton, Highfield, Southampton, United Kingdom
- <sup>8</sup> Civil Engineering and Management, University of Manchester, Manchester, United Kingdom
- <sup>9</sup> Department of Physical Geography, Faculty of Geosciences, Utrecht University, Utrecht, the Netherlands
  - <sup>10</sup> British Geological Survey, Edinburgh, United Kingdom

Correspondence to: Gemma Coxon (gemma.coxon@bristol.ac.uk)

Abstract. Large-sample hydrological datasets containing data for tens to thousands of catchments are invaluable for hydrological process understanding and modelling. CAMELS (Catchment Attributes and MEteorology for Large-sample Studies) datasets provide hydro-meteorological timeseries, catchment attributes and catchment boundaries. Here, we present the second version of CAMELS-GB. CAMELS-GB v2 collates millions of observations from across Great Britain at hourly to monthly timescales, including quality-controlled daily river flows, catchment boundaries, and catchment characteristics from the UK National River Flow Archive. The new features include (1) extended daily hydro-meteorological timeseries from 1970- 2022 including meteorological timeseries from new observed climate datasets, (2) new hourly precipitation, river flow and level timeseries, (3) new groundwater level timeseries and attributes for 55 groundwater wells, and (4) new catchment attributes characterising changing land cover, peak flows and human influences. These data are provided for 671 catchments across Great Britain spanning a diverse range of geophysical characteristics and human influences. CAMELS-GB v2 represents a step change for environmental and modelling analyses across Great Britain, particularly for the characterisation of sub-daily hydrological processes, and is made available as an open dataset (Coxon et al., 2025; https://doi.org/10.5285/9a46d428-958f-4ac1-86eb-94eee70c0955).

## 1 Introduction

Large-sample hydrological (LSH) datasets are invaluable for hydrological process understanding and modelling. LSH datasets provide data for tens to thousands of catchments over national (e.g. Australia, Fowler et al., 2021; Austria, Klingler et al., 2021; Brazil, Almagro et al., 2021; Chagas et al., 2020; Chile, Alvarez-Garreton et al., 2018; Great Britain, Coxon et al., 2020;

50

55

Haiti, Bathelemy et al., 2024; Iceland, Helgason and Nijssen, 2024; Spain, Senent-Aparicio et al., 2024; Sweden, Teutschbein, 2024; Switzerland, Höge et al., 2023; USA, Addor et al., 2017), continental (e.g. Europe, do Nascimento et al., 2024; North America, Arsenault et al., 2020) and global (e.g. CARAVAN, Kratzert et al., 2023; Global Streamflow and Indices and Metadata Archive; Do et al., 2018) scales. While the core data underpinning LSH datasets are streamflow data, LSH datasets also often include meteorological timeseries, simulated timeseries from hydrological models, catchment boundaries and catchment attributes at various spatial and temporal scales. This enables robust benchmarking of hydrological models across a diverse range of catchments (e.g. David et al., 2022; Lees et al., 2021), improved understanding of hydrological processes across environmental gradients (e.g. Addor et al., 2018; Coxon et al., 2024; McMillan et al., 2022), characterisation and prediction of extreme events (e.g. Chagas et al., 2022b; Rasheed et al., 2024) and assessment of the impacts of climate and land management on streamflow (e.g. Chagas et al., 2022a; Slater et al., 2024). Increasingly, LSH datasets have adopted FAIR (finable, accessible, interoperable, and reusable) principles to ensure (1) the large-sample dataset and their source datasets are open-access, and (2) the software tools used to create the large-sample hydrology datasets are open and accessible (Fowler et al., 2025). This has enabled a growing community of LSH datasets, such as the CARAVAN dataset, which included data for 6830 catchments when published in 2023 (Kratzert et al., 2023) but now contains data for more than 20,000 catchments.

CAMELS (Catchment Attributes and MEteorology for Large-sample Studies) datasets are a family of large-sample hydrology datasets that contain hydro-meteorological timeseries, catchment attributes and boundaries for large-samples of catchments for specific countries or regions. CAMELS-GB v1 was the first large-sample, open access dataset for Great Britain (Coxon et al., 2020; Coxon, 2020). It consists of hydro-meteorological catchment time series, catchment attributes (describing topography, climate, hydrology, land cover, soils, hydrogeology, and human influences), and catchment boundaries for 671 catchments. It has been used to understand human impacts on river flows (Bloomfield et al., 2021; Coxon et al., 2024), analyse the spatial sensitivity of river flooding to changes in climate and land cover (Slater et al., 2024), calibrate and evaluate hydrological models (Kiraz et al., 2023) and to benchmark data-driven runoff models (Lees et al., 2021). It has also been incorporated into global catchment datasets (Kratzert et al., 2023). While CAMELS-GB v1 is a valuable dataset, there are important gaps in the current dataset. Firstly, it only contains daily hydro-meteorological timeseries, when sub-daily timeseries are often needed for flood characterisation in small catchments across Great Britain. Secondly, it only contains static catchment attributes (i.e. one snapshot of a geophysical property in time) which makes it challenging to use for trend analyses. Thirdly, groundwater is an important resource in Great Britain, yet there are no timeseries are available for groundwater levels in CAMELS-GB v1.

This paper addresses these data needs by providing a new version of CAMELS-GB. CAMELS-GB v2 contains new datasets including hourly hydro-meteorological timeseries, groundwater level timeseries, dynamic catchment attributes characterising changes in land cover and static catchment attributes characterising groundwater timeseries and reservoirs. We also update the existing data in CAMELS-GB to lengthen the daily hydro-meteorological timeseries and to include the latest rainfall and potential evapotranspiration data for Great Britain. Key differences between the two versions of CAMELS-GB are summarised

https://doi.org/10.5194/essd-2025-608

Preprint. Discussion started: 27 November 2025

© Author(s) 2025. CC BY 4.0 License.

Science Science Data

in Table 1. CAMELS-GB v2 is open access and available on the Environmental Information Data Centre (EIDC). The remainder of the paper describes the changes between v1 and v2, a full description of any new data and advice for users of

CAMELS-GB v2.

70

2 Catchment selection and boundaries

CAMELS-GB v2 contains data for the same 671 catchments as CAMELS-GB v1. These catchments were selected from the UK National River Flow Archive (NRFA) Service Level Agreement (SLA) network (see Dixon et al., 2013; Hannaford, 2004), excluding catchments from Northern Ireland (due to a lack of consistent climate and landscape datasets across the UK) and two gauges where no suitable catchment boundary could be derived. The SLA network ensures a core network of stations for long-term records that undergo additional quality control and validation on the NRFA (Dixon et al., 2013; Muchan and Dixon,

2014). The resulting 671 catchments span a diverse range of hydrological characteristics that represent rivers across GB.

As with CAMELS-GB v1, catchment boundaries are provided in CAMELS-GB v2 as shapefiles in the OSGB 1936 co-ordinate system (British National Grid). The catchment boundaries were derived using the same underlying data and method as for CAMELS-GB v1 (see Section 3 from Coxon et al, 2020 for more details) but updated to better reflect the accuracy of the stations' outlet locations, with minimal influence on the attributes of the final dataset. The catchment boundaries are mostly identical to those provided in CAMELS-GB v1; only two catchments have more than 1% difference in catchment area, with a

maximum difference of 1.5%.

To calculate catchment areal averages for time series and catchment attributes, the exact extract Python package (Baston, 2025) is used to extract data from gridded datasets based on catchment boundary polygons. This tool computes catchment average values by accounting for grid cells that are only partially covered by a polygon. Precisely determining the fractional coverage of each grid cell within a catchment is especially important for small catchments, where coarse approximations can lead to

significant differences in the extracted values.

3 Time Series Data

Daily and hourly hydro-meteorological timeseries are provided for 671 catchments, alongside daily and monthly groundwater level timeseries for 55 boreholes. This section describes the CAMELS-GB v2 timeseries in detail including the source datasets

and differences between products.

3.1 Daily hydro-meteorological timeseries

Daily hydro-meteorological time series are provided for the 671 catchments (see Table 2). The daily time series data includes

key hydro-meteorological variables (streamflow, rainfall, potential evapotranspiration and temperature) from 1st October 1970

3

100

105

- 30<sup>th</sup> September 2022 to provide a valuable, long-term dataset as input and evaluation data for hydrological models, trend analysis and characterisation of hydrological processes.

## 3.1.1 Daily meteorological timeseries

To provide consistency with CAMELS-GB v1, we derive the daily timeseries from the same underlying meteorological datasets; rainfall from CEH Gridded Estimates of Areal Rainfall dataset, potential evapotranspiration and temperature from the Climate hydrology and ecology research support system (CHESS). These datasets were selected due to their high spatial resolution (1 km²), long temporal coverage (>50 years) and basis on the UK climate monitoring network. However, these meteorological datasets are no longer consistently updated and do not cover the full time period required. Consequently, in CAMELS-GB v2 we also provide meteorological timeseries of catchment average rainfall, potential evapotranspiration and temperature from a new UK dataset of gridded climate observations (HadUK-Grid; Hollis et al., 2019) with a national comparison of the different products shown in Figure 1.

Figure 1. National comparison of mean daily a) rainfall (mm day-1) between HadUK-Grid and CEH-GEAR, b) potential evapotranspiration (PET, mm day-1) between Hydro-PE and CHESS-PE PET products, c) potential evapotranspiration with

120

140

interception (PETI, mm day-1) between Hydro-PE and CHESS-PE PETI products and d) temperature (degrees C) between HadUK-Grid and CHESS-met. Mean daily averages calculated from 1st October 1970 – 30th September 2022 for the 671 CAMELS-GB catchments. The blue colours indicate that the HadUK/Hydro-PE daily averages are higher than the CEH-GEAR/CHESS datasets, while the red colours indicate that the HadUK/Hydro-PE daily averages are lower than the CEH-GEAR/CHESS datasets, as a percentage of the CEH-GEAR/CHESS datasets. Contains OS data © Crown Copyright and database right 2025.

Daily rainfall timeseries were derived from two national products; the CEH Gridded Estimates of Areal Rainfall dataset (CEH-GEAR; Keller et al., 2015; Tanguy, 2021) and the HadUK-Grid dataset (Hollis et al., 2019). Both consist of 1 km² gridded estimates of daily rainfall and are based on quality-controlled precipitation data from the Met Office UK rain gauge network. However, the two datasets cover different time periods and the two datasets use different interpolation methods. CEH-GEAR covers 1890 – 2019 whereas HadUK-Grid rainfall is available from 1836 – 2023. The CEH-GEAR dataset uses natural neighbour to interpolate the data, whereas HadUK-Grid uses inverse-distance weighted interpolation to generate the daily rainfall grids. Given the similarities in their underlying datasets, the difference between the two rainfall products is small for most CAMELS-GB catchments (Figure 1a, S1). However, there can be differences of up to 20% in mean annual rainfall totals and larger differences in daily totals for individual catchments (Figure S2).

Daily temperature timeseries were derived from the Climate Hydrology and Ecology research Support System meteorology dataset (CHESS-met; Robinson et al., 2017a) and the HadUK-Grid dataset (Hollis et al., 2019). CHESS-met contains 1 km² gridded estimates of daily mean air temperature (K) from 1961-2019 derived from the Met Office Rainfall and Evaporation Calculation System (MORECS) dataset (Hough and Jones, 1997). MORECS is a 40 km gridded dataset of daily temperature derived from Met Office synoptic stations. For the temperature data in CHESS-met, the MORECS temperature data was interpolated from 40 km resolution to 1km resolution using a bicubic spline and then the temperatures were adjusted to the elevation of each 1km square using the same lapse rate. HadUK-Grid contains 1km² gridded estimates of daily maximum and minimum air temperature (°C) from 1960-2023 derived by interpolating temperature observations from climate observing stations in the Met Office's Integrated Data Archive System (MIDAS). Daily mean temperatures have been calculated by averaging maximum air temperature on day D and minimum air temperature on day D+1 for each day. On average, the difference in mean daily temperature between the two products is relatively small (0.14°C; Figure 1d); however, differences can be larger for individual timesteps (Figure S1, S2).

Daily potential evapotranspiration (PET) timeseries were derived from the Climate Hydrology and Ecology research Support System Potential Evapotranspiration dataset (CHESS-PE; Robinson et al., 2016) and the Hydro-PE HadUK-Grid dataset (Hydro-PE; Brown, 2022; Robinson et al., 2023). Both datasets consist of daily 1km<sup>2</sup> gridded estimates of potential-evapotranspiration for Great Britain calculated using the Penman-Monteith equation for well-watered grass. They also both provide daily potential evapotranspiration with (PETI) and without (PET) an interception correction. Core differences between the datasets are that the PET datasets cover different time periods; 1969-2022 for Hydro-PE and 1961-2019 for CHESS-PE. They also provide different PET estimates, with the Hydro-PE HadUK-Grid dataset providing higher mean annual estimates

of PET (on average 0.1mm day<sup>-1</sup> higher across the CAMELS-GB catchments) and PETI (on average 0.2mm day<sup>-1</sup> higher across the CAMELS-GB catchments (Figure 1b, c). This is due to differences in the underlying data and methodologies used to derive the PET estimates. CHESS-PE is derived from CHESS-met variables (Robinson et al., 2017a) that have been downscaled from the Met Office Rainfall and Evaporation Calculation System (MORECS) dataset (Hough and Jones, 1997), whereas Hydro-PE is derived from HadUK-Grid meteorological data (Hollis et al., 2019). Wind speeds are higher and specific humidity is lower in the HadUK-Grid dataset, and many of the variables in the Hydro-PE HadUK-Grid dataset have been temporally downscaled from monthly to daily using a simple smooth interpolation (for a full discussion of the differences, see Section 5.1 in Robinson et al., 2023). This leads to different estimates of daily PET and PETI between the different datasets across all CAMELS-GB catchments (Figures S1 and S2).

As the meteorological timeseries from HadUK-Grid covers the full time period, we use these data to derive the climate catchment attributes described in Section 5.2.

**Figure 2.** Flow data availability a) length of daily flow timeseries available for each catchment, b) length of hourly flow timeseries available for each catchment, c) percentage of daily and hourly data available for each year from 1970-2022 and 1990-2022. Contains OS data © Crown Copyright and database right 2025.

## 155 3.1.2 Daily hydrological timeseries

Daily streamflow data for the 671 gauges were taken from the UK NRFA on the 7<sup>th</sup> January 2025 (https://nrfaapps.ceh.ac.uk/nrfa/nrfa-api.html, last access 22<sup>nd</sup> January 2025). Streamflow data on the NRFA are provided by

Science Science Discussions

Data

measuring authorities who operate the river flow monitoring network, including the Environment Agency (EA) in England, Natural Resources Wales (NRW) in Wales and Scottish Environmental Protection Agency (SEPA) in Scotland. The streamflow data undergo additional quality control before being uploaded on the NRFA site. Streamflow data in CAMELS-GB v2 are provided as volumetric discharge (m<sup>3</sup> s<sup>-1</sup>) and specific discharge (mm day<sup>-1</sup>).

Figure 2 shows the daily flow data availability for all gauges contained in the CAMELS-GB v2 dataset (Figure 2a) and how this availability changes over time (Figure 2c). Nearly all (666) of the gauges have at least 20 years of daily flow data, and 86 % (577) of the gauges have at least 40 years of daily flow data (Figure 2a). Overall, there is good spatial coverage of long flow time series across Great Britain, with slightly shorter time series concentrated in the north, Midlands and south-east GB. Data availability increases over the time period with 60% of the daily flow data available in 1970, peaking to 99% in the early 2000's and slightly dropping to 96% by 2022 (Figure 2c).

# 3.2 Hourly hydro-meteorological timeseries

Hourly hydro-meteorological time series of rainfall and river flow are provided for the 671 catchments from 1<sup>st</sup> October 1990 09:00 to 1<sup>st</sup> October 2022 08:00 (Table 3). This provides a long-term, high-temporal resolution dataset for model forcing and evaluation, catchment characterisation and analysis of extremes (particularly short-term flood events). Hourly PET is not included in the hourly hydro-meteorological timeseries due to a lack of hourly PET datasets being available and observed hourly climate variables to calculate hourly PET.

# 3.2.1 Hourly meteorological timeseries

Hourly rainfall timeseries are derived from two national products so users have the choice from multiple products. The Gridded Estimates of hourly Areal Rainfall for Great Britain (CEH-GEAR1hr; Lewis et al., 2018, 2022) consists of 1km² gridded estimates of hourly rainfall from 1990-2016. The hourly rainfall estimates are derived from the temporal disaggregation of the CEH-GEAR daily data (described in Section 3.1.1) using hourly gauge data from the Met Office, EA, NRW and SEPA. The hourly gauge data are quality controlled to identify and reject erroneous hourly values in the gauge rainfall input dataset by comparing the gauge data with the CEH-GEAR daily dataset and by implementing a series of quality control tests (Lewis et al., 2018). The nearest neighbour interpolation methodology was used to generate the gridded hourly estimates which were subsequently used to disaggregate the daily data.

Hourly rainfall time series are also derived from the Gauge-Radar Precipitation Dataset (1 hour and 1 km) for Great Britain, GRaD-GB(1H1K), which takes advantage of the accuracy of gauge rainfall and the spatial information of radar rainfall field (Qiu et al., 2025a). The dataset consists of 1km<sup>2</sup> gridded estimates of hourly rainfall for Great Britain from 1 January 2006 to 31 December 2023 and is produced by blending 5-min NIMROD composite radar rainfall (Met Office, 2024) with sub-hourly rainfall observations of ~1800 rain gauge network from the UK Met Office, EA, NRW and SEPA. To produce the hourly rainfall dataset, the radar rainfall and sub-hour rainfall observations are first aggregated to hourly data. Then a quality control

framework is applied to improve the underestimation (radar beam blockage) and overestimation (radar malfunction, ground clutter, and random noise) issues in the radar rainfall data (Qiu et al., 2025b). The quality control procedure that was employed in CEH-GEAR1hr (Lewis et al., 2018) is applied to the gauge rainfall data. A Gauss Blending method was then used to merge the radar rainfall with gauge rainfall. Maintenance work on the rainfall radars means that 3.5% of the hourly timeseries are missing for the CAMELS-GB catchments due to missing radar data. Analysis of GraD-GB(1H1K) shows that the dataset can capture extreme rainfall events missed by rain-gauges (i.e. severe flash flooding in Coverack, Cornwall, 18 July 2017) (Qiu et al., 2025a).

A comparison of the hourly rainfall timeseries from CEH-GEAR1hr and GRaD-GB(1H1K) (Figure 3) shows that on average, GRaD-GB(1H1K) hourly rainfall estimates are 10% higher than CEH-GEAR1hr (range of -25 - 65%) when calculating the average hourly rainfall using the full timeseries. There is more variability between the two products in the north of Great Britain (Figure 3a). The higher average rainfall for the GRaD-GB(1H1K) dataset is partially explained by a higher proportion of wet hours (an hour with >0.1 mm of rainfall) in GRaD-GB(1H1K) (Figure 3b), however, when the average hourly rainfall is calculated using only hours with >0.1 mm of rainfall, the relationship reverses and CEH-GEAR1hr has higher average hourly rainfall despite a lower proportion of wet hours (Figure S3). The CEH-GEAR1hr data are based on gauge data corrected to the daily total and it specifically seeks to preserve intense sub-daily rainfall characteristics in the interpolation method it uses. Therefore, it has fewer wet hours and more intense hourly rainfall than GRaD-GB(1H1K). There is also a relationship with elevation where catchments with lower median elevation have higher average rainfall in the GRaD-GB(1H1K) dataset compared to catchments with higher median elevation (Figure 3c). An example of the differences between the hourly rainfall datasets for individual catchments is shown in Figure S4 and S5.

Figure 3. National comparison of average hourly rainfall (mm hour<sup>-1</sup>) from 1<sup>st</sup> January 2006 to 31<sup>st</sup> December 2016 between CEH-GEAR1hr and the GraD-GB(1H1K) datasets for the 671 CAMELS-GB catchments, a) Difference in average hourly rainfall (average hourly rainfall calculated using all timesteps) (%), b) Fraction of wet hours (wet hours are defined as any hour that recorded rainfall >0.1mm), c) Relationship between median elevation and difference in average hourly rainfall (%). The blue colours indicate that the GRaD-GB(1H1K) hourly rainfall averages are higher than the CEH-GEAR1hr hourly rainfall averages, while the red colours indicate that the GRaD-GB(1H1K) hourly rainfall averages are lower than the CEH-GEAR1hr hourly rainfall averages, as a percentage of the CEH-GEAR1hr dataset. Contains OS data © Crown Copyright and database right 2025

## 3.2.2 Hourly hydrological timeseries

Hourly river flows and levels are provided for 664 and 570 gauges respectively in CAMELS-GB v2 from 1<sup>st</sup> October 1990 – 30<sup>th</sup> September 2022. These hourly river flow and level data combined with the hourly rainfall data provide a wealth of

© Author(s) 2025. CC BY 4.0 License.

additional information beyond the daily data, particularly for flood analyses, convective storm responses, and other short-duration extremes (Figures S4 and S5).

Sub-daily river flows and levels are collected by the measuring authorities. The level data were obtained from SEPA via the timeseries data service (<a href="https://timeseriesdoc.sepa.org.uk/">https://timeseriesdoc.sepa.org.uk/</a>; last access 23<sup>rd</sup> September 2025), from EA primarily through the Hydrology Data Explorer (<a href="https://environment.data.gov.uk/hydrology">https://environment.data.gov.uk/hydrology</a>; last access 23<sup>rd</sup> September 2025) and, where unavailable, with staff assistance, and from NRW entirely with staff assistance. The hourly flow data were obtained from the same sources but derived from the UK-Flow15 dataset (Fileni et al, 2025). UK-Flow15 is a quality-controlled 15-minute flow dataset for the UK, using records from over 1,300 gauging stations including the EA, SEPA and NRW, in addition to the Department for Infrastructure in Northern Ireland and the UK Centre for Ecology & Hydrology. The flow data have been quality controlled using both visual/manual inspection and automated quality control, including novel quality assessment techniques to assess the plausibility of extreme flow events (Fileni et al, 2025).

Here, the 15-minute flow and level data have been aggregated to hourly using a next-hour resample (e.g. 10:00 flow value is the mean of flow recorded at 9:15, 9:30, 9:45 and 10:00). Most gauges (650) have at least 20 years of hourly flow data and 86% (579) of the gauges have at least 30 years of hourly flow data (Figure 2b). There is good spatial coverage of long hourly flow time series across Great Britain, with shorter time series concentrated in the Midlands and south-east of Great Britain. Similar to the daily data, there is greater missing flow data in the earlier part of the record (1990-1995, Figure 2c). There is also good availability of level data where 80% (542) of gauges have at least 20 years of hourly level data and 74% (499) of gauges have at least 30 years of hourly level data.

The flow and level data are also provided with quality control flags. The quality flag for the corresponding hour was selected according to an order of priority (i.e. which flag was deemed to be most important; see Text S1 and Figure S6). No flow or level data have been removed or modified by the quality control process so users can decide which data they want to include as part of their analyses. Users are strongly encouraged to use the flag-based system to identify, remove, or interpolate potentially problematic data as per their study requirements to ensure that only good-quality data are utilised (see Fileni et al, 2025 for a more detailed assessment of the quality control process). The flags are grouped into three categories; (1) comparison with other data products (such as the daily NRFA data), (2) traditional QC checks (such as negative values, truncated low/high flows, spikes), and (3) high-flow QC checks (with comparison to antecedent rainfall and assessment of unrealistically high values) (see Tables S1-3 for a full description of the quality control codes). The levels data underwent less extensive quality control than the flow dataset, and do not include quality control flags for some anomaly checks, comparisons with other UK hydrological products and hydrological similarity flags (see Text S1).

The most common flag for hourly river flow (Figure S8) are where there is a mismatch of >5% between the 15-min values recorded in UK-Flow15 (aggregated to daily) and the National River Flow Archive daily values – this flag is recorded in the timeseries of 539 stations and can appear for over 90% of the timeseries for some gauging stations. The hourly flow data will

not always be consistent with the daily NRFA flow data because of rating curve changes, version control inconsistencies, and station-specific issues.

The other flags typically affect a relatively small proportion of the timeseries (this is expected as many flags focus on extremes) but will be important for users to identify, remove, or interpolate potentially problematic data as per their study requirements.

## 4 Groundwater levels

Groundwater level time series for 55 boreholes across Great Britain were obtained from the British Geological Survey (BGS) (**Figure 4**). These boreholes were selected to fall within CAMELS-GB catchments and represent the main aquifers in Great Britain. Groundwater level data are provided by the measuring authorities (EA, NRW and SEPA) to the National Groundwater Level Archive (NGLA) maintained by the British Geological Survey. Data are measured in meters above Ordnance Datum Newlyn (mAOD), which indicates the groundwater level height at a particular site relative to mean sea level using the national height datum in Britain.

**Figure 4.** Location of 55 groundwater level timeseries in CAMELS-GB-v2. Underlying map shows main aquifers across Great Britain. Devo/Carbo is abbreviated from Devonian/Carboniferous. Contains British Geological Survey materials © UKRI [2025] and OS data © Crown Copyright and database right 2025.

The groundwater level data were provided at irregular time intervals. To provide a standardised dataset for CAMELS-GB-v2, the data were (1) aggregated to monthly by taking an average of all measurements in each calendar month to provide monthly groundwater level timeseries for all boreholes, and (2) also provided at daily timescale where consistent daily or subdaily groundwater level observations were provided. Consequently, monthly groundwater level time series are provided for all 55 boreholes and daily groundwater time series are also provided for 23 of these boreholes. The monthly groundwater timeseries are available for 7 – 72 years with the earliest records beginning in the 1950s. The earliest daily groundwater timeseries start

© Author(s) 2025. CC BY 4.0 License.

Science Science Data

Data

from 1993 and are occasionally averaged from subdaily data. For 23 of the boreholes, outliers in the monthly groundwater level timeseries were flagged as readings 1) diverted from the expected range, or 2) were close to the Datum (see Table S4).

Groundwater well attributes are also provided in CAMELS-GB v2, describing reference and hydrogeological information relating to the wells and boreholes where groundwater level timeseries are provided. For each groundwater well, the name, location (easting and northing), datum, depth and aquifer are provided. The start date, end date and percentage complete of the daily and/or monthly groundwater level timeseries provided in CAMELS-GB v2 are also provided.

## 275 5 Catchment attributes

280

Catchment attributes characterising location, topography, climate, hydrology, land cover, soils, hydrogeology, hydrometry and human influences are provided in CAMELS-GB v2. Most catchment attributes (location, topography, climate, hydrology, soils and hydrogeology) are the same as CAMELS-GB v1 and are re-calculated using the new catchment boundaries or re-extracted from the same source (to account for any changes) (Table 1). Key changes are made for (1) land cover, where changing land cover over multiple years are now provided, (2) hydrometry, where new peak flow information is provided and (3) human influences, where new abstraction, discharge and reservoir attributes are provided (Table 1). This section describes the CAMELS-GB v2 catchment attributes including the source datasets, processing and limitations.

## 5.1 Location, area and topographic data

Catchment attributes describing the location and topography are kept consistent with CAMELS-GB v1 but are re-extracted for each catchment from the NRFA to ensure the latest version of these data. Catchment elevation (min, 10<sup>th</sup>, 50<sup>th</sup>, 90<sup>th</sup>, max) within the catchment mask is derived from CEH's 50m Integrated Hydrological Digital Terrain Model. Mean elevation and mean drainage path slope (index of catchment steepness) are also provided based on methods developed for the Flood Estimation Handbook (Bayliss, 1999), except for two catchments (18011 and 26006) where catchment boundaries could not be automatically derived. For more information see Section 6.1 of Coxon et al., (2020).

#### 290 5.2 Climatic indices

The same suite of climatic indices is calculated as CAMELS-GB v1 (and other CAMELS datasets). These climatic indices characterise long-term (i.e. mean precipitation and PET, aridity index), seasonal (i.e. rainfall seasonality and fraction of snow), and short-term (i.e. frequency, duration and timing of high and low precipitation events) climate dynamics. The climatic indices in CAMELS-GB v2 are derived using the HadUK-Grid catchment daily rainfall, potential evapotranspiration (without correction for interception), and temperature time series described in Section 3.1.1 of this paper. These data were chosen (rather than the CEH-GEAR and CHESS-PE data) as the HadUK-Grid data cover the full timeseries available.

# 5.3 Hydrologic signatures

The same suite of hydrologic signatures is calculated as CAMELS-GB v1 (and other CAMELS datasets). The hydrologic signatures characterise long-term (i.e. mean flow, runoff ratio, streamflow elasticity, baseflow index), seasonal (i.e. the half low date), and short-term (i.e. high and low flow percentiles, frequency and duration of high and low streamflow events) streamflow dynamics. The hydrologic signatures in CAMELS-GB v2 are derived using the HadUK-Grid catchment daily rainfall and streamflow time series from 1 October 1970 to 30 September 2022 (Section 3.1.1 and 3.1.2). Users should consider the availability of streamflow data (i.e. length and percentage missing) when comparing hydrologic signatures across catchments (McMillan et al, 2023).

#### 5.4 Land cover attributes

Land cover for multiple years is provided in CAMELS-GB v2. Land cover attributes for each catchment were derived from the UK Land Cover Map 1990, 2015, 2017, 2018, 2019, 2020, 2021 and 2022 produced by UK CEH (Marston et al., 2022, 2024; Morton et al., 2020a, b, c, 2021; Rowland et al., 2017, 2020). While there are also land cover maps produced by UK CEH for 2000 and 2007, these do not use a consistent methodology for derivation of their land cover classes, preventing straightforward analysis of changes in land cover over time.

The land cover maps are derived by classifying satellite imagery into 21 classes based on the Joint Nature Conservation Committee (JNCC) broad habitats. Like CAMELS-GB v1, these 21 classes are mapped to eight land cover classes that describe deciduous woodland, evergreen woodland, grass and pasture, shrubs, crops, suburban and urban, inland water, bare soil, and rocks. For CAMELS-GB v2, we use the 25 m raster data from the range of LCM products consisting of the most likely land cover type for each grid cell. For each catchment, the percentage of the catchment covered by each of the eight land cover types was calculated and is provided in CAMELS-GB v2 for 1990, 2015, 2017, 2018, 2019, 2020, 2021 and 2022 (Table 5).

Users should be aware that while advances in methods mean land cover is more consistent between years, there are still issues around accuracy, stability between years and the strength of the change signal. This means that while some of the land cover change between years is real, there is also noise in the variability between years. This could be due to (1) the timing and quality of satellite imagery, particularly in high elevation catchments where there is often limited satellite imagery due to snowy/cloudy conditions, (2) differences in methods applied between years and (3) confusion in land cover classes (Rowland et al., 2024). As an example, we show changes in urban land cover for catchments in CAMELS-GB v2 in Figure S9. While a trend of increasing land cover over time is apparent for many catchments, there is variability between years and most catchments experience a decrease in urban cover from 2021-2022 that is unlikely to be reflected in the real world.

### 325 **5.5** Soil attributes

Soil attributes are the same as CAMELS-GB v1 but derived using the updated catchment boundaries described in Section 2. A brief summary of the underlying data is provided here, with a full summary provided in Section 6.5 of Coxon et al., (2020).

Soil attributes of depth available to roots, percentage sand, silt and clay content, organic carbon content, bulk density, and total available water content for each catchment were calculated from the European Soil Database Derived Data product (Hiederer, 2013a, b). Saturated hydraulic conductivity and porosity (saturated volumetric water content) were also estimated from these variables using two pedo-transfer functions (based on Cosby et al, 1984 and Wosten et al, 1999, 2000, 2001). A weighted mean of the topsoil and subsoil data was calculated for all 1km grid cells and then used to calculate average soil properties for each catchment either by calculating the arithmetic mean or harmonic mean (for saturated hydraulic conductivity, Samaniego et al., 2010) of all 1km grid cells within the catchment boundary. To capture the spatial heterogeneity and data availability of the soils data, the 5th, 50th, and 95th percentile and percentage of no-data values of all grid cell values falling within the catchment boundaries are also provided.

# 5.6 Hydrogeological attributes

Hydrogeological attributes are the same as CAMELS-GB v1 but derived using the updated catchment boundaries described in Section 2. A brief summary of the underlying data is provided here, with a full summary provided in Section 6.6 of Coxon et al., (2020).

Hydrogeological attributes were derived from products produced by the British Geological Survey; the UK bedrock hydrogeological map (BGS, 2019) and a superficial deposit productivity map. These two datasets were merged to categorise the uppermost geological layer into nine classes that account for superficial deposits (where present) and bedrock (where superficial deposits are absent). The nine classes indicate how hydrogeology affects river flow behaviour by characterising the proportion of the catchment that is covered by deposits that have high, moderate, or low productivity, and whether the dominant water flow is through fractures or between grains (Table 5).

## 5.7 Hydrometry and discharge uncertainty attributes

The hydrometry and discharge uncertainty attributes describe the gauging station type (i.e. the type of weir, structure, or measurement device used to measure flows), period of flow data available (i.e. start date, end date and percentage complete), gauging station discharge uncertainty, peak flow information (i.e. the maximum gauged flow and percentage of extrapolation and channel characteristics (such as bankfull) (see Table 5). Many of these attributes are the same as CAMELS-GB v1. The gauging station type and channel characteristics were re-extracted from the NRFA with changes for a small number of stations where the gauging station type has changed since CAMELS-GB v1. The period of flow data were updated for the daily timeseries and new catchment attributes added to describe the period of hourly flow data. The gauging station discharge

uncertainties were calculated using the same method from Coxon et al., (2015) but updated so that discharge uncertainties were calculated from flow percentiles from the longer streamflow timeseries. A full description of the discharge uncertainty attributes is provided in Section 6.7 of Coxon et al., (2020).

New hydrometry attributes were added describing peak flow information; including the (1) maximum gauging flow (the highest manual measurement of flow taken at a gauging station) and the date this maximum gauging flow was taken, (2) the maximum daily/hourly flow recorded in the catchment timeseries in CAMELS-GB v2, and (3) the percentage of time (excluding NaNs) that the daily/hourly flow timeseries in CAMELS-GB v2 is higher than the maximum gauging flow. Figure 5 shows that for most gauges (70% for daily and 69% for hourly), the flow is extrapolated above the maximum flow gauging for a small proportion (

Figure 5. Comparison of the maximum gauging flow with a) maximum daily flow and b) maximum hourly flow recorded in the catchment timeseries in CAMELS-GB v2, where dots are coloured by the percentage of time (excluding NaNs) that the daily/hourly flow timeseries in CAMELS-GB v2 is higher than the maximum gauging flow. c) ratio of maximum flow gauging to maximum daily/hourly flow in flow timeseries in CAMELS-GB v2.

## 5.8 Human influence attributes

CAMELS-GB contains many catchments impacted by human activities, so we aim to provide attributes that help users quantify and characterise human influences in each of the catchments. New, open-source datasets are used to quantify average abstraction and discharge in each catchment, and we provide new reservoir attributes characterising the size and location of the reservoirs relative to the gauge.

## 5.8.1 Benchmark catchments

The UK Benchmark Network contains 146 catchments where human impacts on flow regimes are assumed to be minimal (Harrigan et al., 2018). All CAMELS-GB catchments are identified as either being part of this network or not to provide users

© Author(s) 2025. CC BY 4.0 License.

390

Science Science Data

with an indication of 'near-natural' catchments and suitable for studies where human impacts need to be minimal. Users should be aware that to ensure coverage in the south and east of GB (where there are lots of human influences on river flows), some human impacts were accepted.

## 5.8.2 Abstraction and discharges

Average daily abstraction and discharge rates are provided again in CAMELS-GB v2 but based on a new dataset of abstractions and discharges (Rameshwaran et al., 2025). This new dataset is based the same underlying data as used in CAMELS-GB v1 but underwent additional quality control and is now available open source.

The abstraction data consist of the total water quantity (in most cases measured using a water meter) that has been abstracted for each license and each month from 1999 to 2014 in England on a 1km grid. These monthly abstraction data were averaged to provide a mean monthly abstraction from 1999 to 2014 for each abstraction licence and then aggregated for each catchment to provide a mean daily abstraction rate for all English catchments in CAMELS-GB v2 for groundwater and surface water sources. The use of the abstracted water (agriculture, amenities, environmental, industrial, energy, or water supply) is also provided and how much of the abstracted water is consumed/lost (high- 100%, medium – 60%, low – 3% and very low – 0.3%). For example, cress pond throughflow is described as very low loss, whilst farming and water supply is classed as medium loss, and trickle irrigation is classed as high. These loss factors are only used for billing purposes and therefore indicative of the true water consumption.

The discharges data consist of recent actual discharges for England from the WRGIS (Water Resources Geographic Information System). These data represent discharges from sewage treatment works and other 'significant' discharges (typically those >20 m³ day¹) using an estimate of recent actual summer discharge. For each catchment, we calculate a sum of all the discharges that fall within the catchment boundary and then convert into millimetres per day using catchment area to provide a mean daily discharges rate.

There are several limitations associated with these data. Firstly, these catchment attributes are only available for England and there are many catchments where either (1) no data are available (identified by "NaN"), (2) abstractions or discharges are recorded in zero when in reality they are not, or (3) only a proportion of the abstractions/discharges are accounted for, as the catchments lie on the border of England–Wales or England–Scotland. Secondly, the topographical catchment mask was used to define which abstraction returns were included in each catchment which will not be representative for groundwater abstractions that lie within the topographical catchment but do not have a direct impact on the catchment streamflow or those that lie outside the catchment but have an impact on that catchment's streamflow. Thirdly, this is not the full picture of human influences within each catchment. Not all licence types/holders are required to submit records to the Environment Agency; the abstraction data used here does not hold returns for abstractions less than 20 m³ day⁻¹, and from 2008, abstraction licence-holders less than 100 m³ day⁻¹ were no longer required to submit records of abstraction. Furthermore, there is large inter-annual

and intra-annual variation in the abstraction and discharge data, and its impacts will be different across the flow regime. Finally, while abstractions represent water removed from surface water or groundwater sources, some of this water will be returned to catchment storages. This is partially represented by the loss factor and discharges information but the relationship between discharge consent data and water returned from abstractions will often be more complex than these simple attributes.

#### 5.8.3 Reservoirs

For CAMELS-GB v2, several reservoir attributes are derived for each catchment by determining the reservoirs that lie within the catchment mask from the reservoir locations and then calculating (1) the number of reservoirs in each catchment; (2) their combined capacity; (3) the fraction of that capacity that is used for hydroelectricity, navigation, drainage, water supply, flood storage, and environmental purposes; (4) the year when the first and last reservoirs in the catchment were built, and (5) the contributing area and normalised upstream capacity.

The first four sets of reservoir attributes are the same as CAMELS-GB v1 and calculated from the open-source UK reservoir inventory (Durant and Counsell, 2018) supplemented with information from SEPA's publicly available controlled reservoirs register. For CAMELS-GB v2, we excluded 43 reservoirs from the inventory as they could not be placed on the river network largely because their outflow or inflow location was unclear (see Figure S1 in Salwey et al., 2023) and therefore the new reservoir attributes could not be calculated. This leads to only minor differences with CAMELS-GB v1 (Figure S10).

Two new reservoir attributes are included in CAMELS-GB v2; contributing area and normalised upstream capacity. These attributes were chosen due to previous studies finding clear links between the size and location of upstream reservoirs and the associated flow alteration for UK catchments (Salwey et al., 2023).

Contributing area describes the percentage of the overall catchment surface area that is drained through reservoirs:

Contributing area (%) = 
$$\frac{\text{catchment area drained by reservoirs } (km^2)}{\text{total catchment area } (km^2)} \times 100$$

In CAMELS-GB v2, the mean contributing area is 18%, with a maximum of 100%. The contributing area is complemented by the normalised upstream capacity, which compares the capacity of a reservoir to the average volume of precipitation received by the catchment in a year:

Normalised upstream capacity

$$= \frac{total\ upstream\ reservoir\ capacity\ (mm^3)}{total\ catchment\ area\ (mm^2)*average\ annual\ catchment\ precipitation\ (mm)}$$

In CAMELS-GB v2, the average normalised upstream capacity is 0.08 (i.e. the reservoir is large enough to store 8% of average annual rainfall) with a maximum of 2.5 (i.e. the reservoir is large enough to store 250% of average annual rainfall). Ten catchments have a normalised upstream capacity greater than 0.25.

# 6 Data availability

The CAMELS-GB v2 dataset available under an Open Government License via the UK Centre for Ecology & Hydrology Environmental Information Data Centre (Coxon et al., 2025; <a href="https://doi.org/10.5285/9a46d428-958f-4ac1-86eb-94eee70c0955">https://doi.org/10.5285/9a46d428-958f-4ac1-86eb-94eee70c0955</a>). The data contain catchment boundaries, hydro-meteorological and groundwater time series (at hourly, daily and monthly time-scales), catchment attributes and groundwater well attributes as described above. The data format is described in the supporting documentation available on the UK Centre for Ecology & Hydrology Environmental Information Data Centre.

## 7 Code availability

The exact extract Python package (Baston, 2025) is used to extract catchment average data from gridded datasets based on the catchment boundary polygons described in Section 2. The code from <a href="https://github.com/naddor/camels">https://github.com/naddor/camels</a> (last access: 23<sup>rd</sup> September 2025) was used to generate the climatic indices and hydrological signatures.

## 8 Conclusions

This paper presents the second version of CAMELS-GB. CAMELS-GB v2 collates millions of observations from across Great Britain at hourly to monthly timescales, including quality-controlled daily river flows, catchment boundaries, and catchment characteristics from the UK National River Flow Archive. The new features include (1) extended daily hydro-meteorological timeseries up to 2022 including meteorological timeseries from new observed climate datasets, (2) new hourly precipitation, river flows and level timeseries, (3) new groundwater level timeseries and attributes for 55 groundwater wells, and (4) new catchment attributes characterising changing land cover, peak flows and human influences.

CAMELS-GB v2 provides exciting new opportunities for environmental and modelling analyses across Great Britain. This includes enabling the development of common frameworks for model evaluation and benchmarking at regional to national scales and the analysis of hydrologic variability across the UK. The new sub-daily hydro-meteorological timeseries provide a wealth of additional information beyond the daily data, particularly for flood analyses, convective storm responses, and other short-duration extremes. The new catchment attributes enable users to explore how different catchment characteristics control river flow behaviour, particularly in human-influenced catchments. Future updates to the dataset will concentrate on greater spatial and temporal coverage of the groundwater level data and river network characteristics (e.g. Strahler Index).

475

Science Science Data

## 9 Author Contribution

GC led and produced CAMELS-GB v2 with the following contributions: (1) YZ derived all the catchment timeseries data from gridded datasets, (2) FF produced the hourly flow and level timeseries with contributions from HF, EL, MF, HC and HH, (3) XQ produced the GRaD-GB(1H1K) with contributions from HF, AG and EL, (4) MF and RB provided the daily streamflow data, catchment boundaries and all catchment attributes sourced from the National River Flow Archive, (5) TG provided the groundwater level timeseries, DEW processed these timeseries and produced the groundwater attributes, (6) GN derived the soils and abstraction attributes, (7) SS derived the climatic attributes, hydrologic signatures, hydrogeology attributes and the reservoir attributes, (8) HC facilitated the upload of the dataset to the EIDC. All co-authors contributed to the design of the dataset. The manuscript was prepared by GC with contributions from all co-authors.

# 480 10 Acknowledgments

The authors would like to express their great appreciation to all the data collectors, processors, and providers who made this work possible, particularly at the UK Centre for Ecology & Hydrology, the National River Flow Archive, UK Met Office, Environment Agency, Natural Resources Wales, Scottish Environmental Protection Agency, British Geological Survey and the National Groundwater Level Archive. In particular, many thanks to the staff from the Environment Agency, Natural Resources Wales and Scottish Environmental Protection Agency involved in the provision of sub-daily flow data.

Many thanks to Professor Louise Slater (University of Oxford) and Clare Rowland (UK Centre for Ecology & Hydrology) who provided valuable advice on the land cover data. Many thanks also to Dr Rosanna Lane (UK Centre for Ecology & Hydrology) for valuable advice on the soil catchment attributes.

This work was supported by the Natural Environment Research Council (NERC-UKRI) as part of the Floods and Droughts
Research Infrastructure (FDRI) project. The views and opinions expressed are those of the authors alone. NERC-UKRI is not responsible for any application of the data/information.

This work was made possible through the Environment Agency Flood Hydrology Improvement Programme.

DW and TG publish with the permission of the Director, British Geological Survey (UKRI).

# 11 Financial Support

GC, YZ, DEW and GN were supported by a United Kingdom Research and Innovation Future Leaders Fellowship (MR/V022857/1). GC and YZ were also supported by the Natural Environmental Research Council Large Grant SMARTWATER (NE/X018865/1). SS was funded by the DAFNI Centre of Excellence for Resilient Infrastructure Analysis

(ST/Y003713/1) and the European Union (ERC, MultiDry, Grant Agreement number: 101075354). FF was funded by a NERC sponsored ONE Planet Doctoral Training Partnership (NE/S007512/1).

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

Tables

Table 1 Summary of changes to CAMELS-GB dataset for version 2

| Data                    |                                               | Data provided in                                                                                                                                                                                | Change for CAMELS-GB v2                                                                                                                                                                                                                                                                                                                                                                | Section |
|-------------------------|-----------------------------------------------|-------------------------------------------------------------------------------------------------------------------------------------------------------------------------------------------------|----------------------------------------------------------------------------------------------------------------------------------------------------------------------------------------------------------------------------------------------------------------------------------------------------------------------------------------------------------------------------------------|---------|
|                         |                                               | CAMELS-GB v1                                                                                                                                                                                    |                                                                                                                                                                                                                                                                                                                                                                                        |         |
| Catchments              |                                               | 671 catchments across Great Britain                                                                                                                                                             | No change to number of catchments or catchment selection. Catchment boundaries updated.                                                                                                                                                                                                                                                                                                | 2       |
| Timeseries              | Daily hydro-<br>meteorological<br>timeseries  | Timeseries available from 1st October 1970 - 30th September 2015 for streamflow, rainfall, potential evapotranspiration, temperature, wind speed, humidity, short-wave and long-wave radiation. | New timeseries data for rainfall, PET and temperature from the HadUK dataset to provide multiple estimates of climatic variables with an extended timeseries up to 30th September 2022. CEH-GEAR rainfall and CHESS timeseries are extended to 30th September 2019.  No longer providing wind speed, humidity, short-wave radiation and long-wave radiation as these were rarely used. | 3.1     |
|                         | Hourly hydro-<br>meteorological<br>timeseries | None provided                                                                                                                                                                                   | Hourly streamflow, river level and rainfall (from two products) from 1st October 1990 – 30th September 2022.                                                                                                                                                                                                                                                                           | 3.2     |
|                         | Groundwater<br>level timeseries               | None provided                                                                                                                                                                                   | Daily groundwater level timeseries for 23 wells from 1993-2025 and monthly groundwater level timeseries for 55 wells from 1952-2025.                                                                                                                                                                                                                                                   | 4       |
| Catchment<br>Attributes | Location and Topography                       | Hydrometry attributes included gauge ID, location, catchment area and elevation percentiles                                                                                                     | The same attributes are provided but updated to the latest information from the UK National River Flow Archive.                                                                                                                                                                                                                                                                        | 5.1     |
|                         | Climatic                                      | Climatic indices including mean rainfall and PET,                                                                                                                                               | The same climatic indices are provided but calculated using the                                                                                                                                                                                                                                                                                                                        | 5.2     |

|              | aridity index, seasonality,  | extended Had-UK daily rainfall,    |     |
|--------------|------------------------------|------------------------------------|-----|
|              | snow fraction and            | temperature and PET timeseries.    |     |
|              | frequency, duration and      |                                    |     |
|              | timing of climatic extremes  |                                    |     |
|              | were provided for each       |                                    |     |
|              | catchment. Derived using     |                                    |     |
|              | daily meteorological         |                                    |     |
|              | timeseries.                  |                                    |     |
| Hydrologic   | Hydrologic signatures        | The same hydrologic signatures     | 5.3 |
|              | including mean streamflow,   | are provided but calculated using  |     |
|              | runoff ratio, slope of the   | the extended Had-UK daily          |     |
|              | flow duration curve,         | rainfall and daily streamflow      |     |
|              | baseflow index, frequency,   | timeseries.                        |     |
|              | duration and timing of       |                                    |     |
|              | low/high flow events were    |                                    |     |
|              | provided for each            |                                    |     |
|              | catchment. Derived using     |                                    |     |
|              | daily hydro-meteorological   |                                    |     |
|              | timeseries.                  |                                    |     |
|              | timeseries.                  |                                    |     |
| Land Cover   | Land cover percentages for   | Land cover for multiple years is   | 5.4 |
|              | eight land cover classes     | provided including 1990, 2015,     |     |
|              | provided for each            | 2017, 2018, 2019, 2020, 2021 and   |     |
|              | catchment for a single year  | 2022. Same land cover classes are  |     |
|              | (2015).                      | provided.                          |     |
| Soils        | Soil attributes              | The same soil attributes are re-   | 5.5 |
|              | characterising the soil      | calculated using the same          |     |
|              | texture, porosity, saturated | underlying data but with the new   |     |
|              | conductivity and depth.      | catchment boundaries (very little  |     |
|              | _                            | change).                           |     |
| Hydrogeology | Hydrogeological attributes   | The same hydro-geological          | 5.6 |
|              | characterising the upper     | attributes are re-calculated using |     |
|              | geological layer describing  | the same underlying data but with  |     |
|              | the proportion of the        | the new catchment boundaries       |     |
|              | catchment covered by         | (very little change).              |     |
|              | deposits of high, moderate,  | , ,                                |     |
|              | or low productivity and      |                                    |     |
|              | whether this is              |                                    |     |
|              | whether this is              |                                    |     |

|             | 7           | 1                            |                                     |     |
|-------------|-------------|------------------------------|-------------------------------------|-----|
|             |             | predominantly via fracture   |                                     |     |
|             |             | or intergranular flow.       |                                     |     |
|             | YY 1        | XX 1                         | 771                                 |     |
|             | Hydrometry  | Hydrometry and discharge     | The same attributes are provided    | 5.7 |
|             |             | uncertainty attributes       | but updated to latest information   |     |
|             |             | describing the gauging       | from the UK National River Flow     |     |
|             |             | station type, period of flow | Archive. New hydrometry             |     |
|             |             | data available, gauging      | attributes are provided to          |     |
|             |             | station discharge            | characterise peak flow              |     |
|             |             | uncertainty and channel      | information.                        |     |
|             |             | characteristics were         |                                     |     |
|             |             |                              |                                     |     |
|             |             | 1                            |                                     |     |
|             |             | catchment.                   |                                     |     |
|             | Human       | Human influence attributes   | Abstractions and discharges are     | 5.8 |
|             | Influences  | describing abstractions,     | based on a new open-access          |     |
|             |             | discharges and reservoir     | dataset and new attributes added    |     |
|             |             | attributes were provided for | describing the loss factor of       |     |
|             |             | each catchment.              | abstractions. New attributes added  |     |
|             |             |                              | describing the reservoir            |     |
|             |             |                              | contributing area and normalised    |     |
|             |             |                              | upstream capacity.                  |     |
|             |             |                              |                                     |     |
| Groundwater | Groundwater | None provided                | Groundwater well attributes are     | 4   |
| well        | Wells       |                              | provided, describing reference      |     |
| attributes  |             |                              | and hydrogeological information     |     |
|             |             |                              | relating to the wells and boreholes |     |
|             |             |                              | where groundwater level             |     |
|             |             |                              | timeseries are provided.            |     |
|             |             |                              | timeseries are provided.            |     |

 $\textbf{Table 2} \ \textbf{Summary table of catchment daily hydro-meteorological time series available in CAMELS-GB v2}$ 

| Timeseries<br>Class | Timeseries Name       | Description              | Unit  | Data Source         |
|---------------------|-----------------------|--------------------------|-------|---------------------|
| Meteorological      | precipitation_cehgear | catchment averaged daily | mm    | CEH-GEAR (Keller et |
| Timeseries          |                       | precipitation            | day-1 | al., 2015; Tanguy,  |
| (available from     |                       |                          |       | 2021)               |

| 1st October 1970                                                          | precipitation haduk           | catchment averaged daily                                                                                                                                                                                 | mm                      | HadUK-Grid (Hollis                                          |
|---------------------------------------------------------------------------|-------------------------------|----------------------------------------------------------------------------------------------------------------------------------------------------------------------------------------------------------|-------------------------|-------------------------------------------------------------|
| - 30 <sup>th</sup>                                                        | precipitation_naduk           | precipitation                                                                                                                                                                                            | day-1                   | et al., 2019)                                               |
|                                                                           |                               |                                                                                                                                                                                                          | uay                     | Ct al., 2019)                                               |
| September<br>2022)                                                        | pet_chess                     | catchment averaged daily<br>potential evapotranspiration<br>for a well-watered grass<br>(Penman-Monteith equation)                                                                                       | mm<br>day <sup>-1</sup> |                                                             |
|                                                                           | peti_chess                    | catchment averaged daily potential evapotranspiration for a well-watered grass (Penman-Monteith equation with a correction added for interception on days where rainfall has occurred)                   | mm<br>day <sup>-1</sup> | CHESS-PE (Robinson et al., 2017b, 2023b)                    |
|                                                                           | pet_hydrope                   | catchment averaged daily<br>potential evapotranspiration<br>for a well-watered grass<br>(Penman-Monteith equation)                                                                                       | mm<br>day <sup>-1</sup> |                                                             |
|                                                                           | peti_hydrope                  | catchment averaged daily<br>potential evapotranspiration<br>for a well-watered grass<br>(Penman-Monteith equation<br>with a correction added for<br>interception on days where<br>rainfall has occurred) | mm<br>day <sup>-1</sup> | Hydro-PE (Brown et<br>al., 2023; Robinson et<br>al., 2023c) |
|                                                                           | temperature_chess             | catchment averaged daily temperature                                                                                                                                                                     | °C                      | CHESS-met<br>(Robinson et al.,<br>2023a)                    |
|                                                                           | temperature_haduk             | catchment averaged daily temperature                                                                                                                                                                     | °C                      | HadUK-Grid (Hollis<br>et al., 2019)                         |
| Hydrological Timeseries (available from 1st October 1970 - 30th September | discharge_spec  discharge_vol | catchment specific discharge<br>(converted to mm day-1 using<br>catchment areas described in<br>Section 2)                                                                                               | mm day-1                | UK National River Flow Archive using the NRFA API*          |
| 2022)                                                                     |                               |                                                                                                                                                                                                          |                         |                                                             |

 $\textbf{Table 3} \ \textbf{Summary table of catchment hourly hydro-meteorological time series available in CAMELS-GB v2}$ 

| Timeseries                                                                               | Timeseries                    | D                                                                                                                                                                                                                                                                                                                         | TT **                                    | D + C                                                                                                                                                                                                                                                                                                                                   |
|------------------------------------------------------------------------------------------|-------------------------------|---------------------------------------------------------------------------------------------------------------------------------------------------------------------------------------------------------------------------------------------------------------------------------------------------------------------------|------------------------------------------|-----------------------------------------------------------------------------------------------------------------------------------------------------------------------------------------------------------------------------------------------------------------------------------------------------------------------------------------|
| Class                                                                                    | Name                          | Description                                                                                                                                                                                                                                                                                                               | Unit                                     | Data Source                                                                                                                                                                                                                                                                                                                             |
| Meteorological Timeseries (available from                                                | precipitation_cehgear         | catchment hourly averaged precipitation from 1st October 1990 - 31st December 2019                                                                                                                                                                                                                                        | mm hour-1                                | CEH-GEAR1hr (Lewis et al., 2018, 2022)  Gauge-Radar                                                                                                                                                                                                                                                                                     |
| 1st October 1990<br>09:00 – 1st<br>October 2022<br>08:00)                                | gradgb                        | precipitation from 1st January<br>2006 - 31st December 2023                                                                                                                                                                                                                                                               |                                          | Precipitation Dataset (1 hour and 1 km) for Great Britain, GRaD-GB(1H1K) (Qiu et al., 2025b, a)                                                                                                                                                                                                                                         |
|                                                                                          | discharge_spe c discharge_vol | catchment specific discharge (converted to mm hour-1 using catchment areas described in Section 2)  catchment discharge                                                                                                                                                                                                   | mm hour <sup>-1</sup> m³ s <sup>-1</sup> | The flows and level data were obtained from SEPA via the timeseries data service (https://timeseriesdoc.s epa.org.uk/; last access                                                                                                                                                                                                      |
| Hydrological Timeseries (available from 1st October 1990 09:00 – 1st October 2022 08:00) | discharge_fla g level         | numeric flag that indicates the quality of the flow data (full description of flags can be found in Figure S5 and Table S1-S3)  height of the river (measured in metres above river bed)  numeric flag that indicates the quality of the level data (full description of flags can be found in Figure S5 and Table S1-S3) | m -                                      | epa.org.uk/; last access 23rd September 2025), from EA primarily through the Hydrology Data Explorer (https://environment.da ta.gov.uk/hydrology; last access 23rd September 2025) and, where unavailable, with staff assistance, and from NRW entirely with staff assistance. Flags were derived from UKFlow-15 (Fileni et al., 2025). |

<sup>\*</sup> https://nrfaapps.ceh.ac.uk/nrfa/nrfa-api.html, data downloaded on the  $7^{th}$  January 2025, last access to website  $23^{rd}$  September 2025

 $\textbf{Table 4.} \ \textbf{Summary table of daily and monthly groundwater level timeseries available in CAMELS-GB v2}$ 

| Timeseries<br>Class | Timeseries<br>Name | Description     | Unit | Data Source                               |
|---------------------|--------------------|-----------------|------|-------------------------------------------|
| Groundwat           | groundwater_le     | groundwater     | mAOD | British Geological Survey (BGS), National |
| er Level            | vel                | level for a     |      | Groundwater Level Archive [online].       |
| Timeseries          |                    | specific        |      | Available at:                             |
| (daily and          |                    | borehole at     |      | https://www2.bgs.ac.uk/groundwater/datai  |
| monthly             |                    | either daily or |      | nfo/levels/ngla.html (Accessed: 01 Feb    |
| timeseries          |                    | monthly         |      | 2025).                                    |
| for variable        |                    | timesteps.      |      |                                           |
| time periods)       |                    |                 |      |                                           |

**Table 5.** Summary table of attributes available in CAMELS-GB v2

| Attribute                                             | Attribute      | Description.                                                                                                                                                                                              | 11:4                 | Data Samua                                                                                          |
|-------------------------------------------------------|----------------|-----------------------------------------------------------------------------------------------------------------------------------------------------------------------------------------------------------|----------------------|-----------------------------------------------------------------------------------------------------|
| Class                                                 | Name           | Description                                                                                                                                                                                               | Unit                 | Data Source                                                                                         |
|                                                       | gauge_id       | catchment identifier (corresponds to the gauging station ID provided by the NRFA)                                                                                                                         | -                    |                                                                                                     |
|                                                       | gauge_name     | gauge name (river name followed by gauging station name)                                                                                                                                                  | -                    | UK National<br>River Flow                                                                           |
|                                                       | gauge_lat      | gauge latitude                                                                                                                                                                                            | 0                    | Archive using                                                                                       |
|                                                       | gauge_lon      | gauge longitude                                                                                                                                                                                           | 0                    | the NRFA API*                                                                                       |
|                                                       | gauge_easting  | gauge easting                                                                                                                                                                                             | m                    |                                                                                                     |
| Location and                                          | gauge_northing | gauge northing                                                                                                                                                                                            | m                    |                                                                                                     |
| Topography                                            | gauge_elev     | gauge elevation                                                                                                                                                                                           | m.a.s.l              |                                                                                                     |
|                                                       | area           | catchment area                                                                                                                                                                                            | km <sup>2</sup>      | CEH's                                                                                               |
|                                                       | dpsbar         | catchment mean drainage path slope                                                                                                                                                                        | m km <sup>-1</sup>   | Integrated                                                                                          |
|                                                       | elev_mean      | catchment mean elevation                                                                                                                                                                                  | m.a.s.l              | Hydrological                                                                                        |
|                                                       | elev_min       | catchment minimum elevation                                                                                                                                                                               | m.a.s.l              | Digital Terrain                                                                                     |
|                                                       | elev_10        | catchment 10 <sup>th</sup> percentile elevation                                                                                                                                                           | m.a.s.l              | Model (Morris                                                                                       |
|                                                       | elev_50        | catchment median elevation                                                                                                                                                                                | m.a.s.l              | and Flavin,                                                                                         |
|                                                       | elev_90        | catchment 90 <sup>th</sup> percentile elevation                                                                                                                                                           | m.a.s.l              | 1990)                                                                                               |
|                                                       | elev_max       | catchment maximum elevation                                                                                                                                                                               | m.a.s.l              |                                                                                                     |
|                                                       | p_mean         | mean daily precipitation                                                                                                                                                                                  | mm day-1             | Catchment                                                                                           |
| Climatic                                              | pet_mean       | mean daily PET (Penman-Monteith equation without interception correction)                                                                                                                                 | mm day <sup>-1</sup> | timeseries of precipitation,                                                                        |
| Indices (computed for 1st Oct 1970 to 30th Sept 2022) | aridity        | aridity, calculated as the ratio of mean daily potential evapotranspiration to mean daily precipitation                                                                                                   | -                    | precipitation, potential evapotranspirati on and temperature described in Section 3.1.1 and Table 2 |
|                                                       | p_seasonality  | seasonality and timing of precipitation (estimated using sine curves to represent the annual temperature and precipitation cycles; positive (negative) values indicate that precipitation peaks in summer | -                    |                                                                                                     |

|                                       | frac_snow high_prec_freq high_prec_dur                        | (winter) and values close to zero indicate uniform precipitation throughout the year)  fraction of precipitation falling as snow (for days colder than $0^{\circ}$ C)  frequency of high precipitation days ( $\geq 5$ times mean daily precipitation)  average duration of high precipitation events (number of consecutive days $\geq 5$ times mean daily precipitation)                                     | - days yr-1 days     |                                                                 |
|---------------------------------------|---------------------------------------------------------------|----------------------------------------------------------------------------------------------------------------------------------------------------------------------------------------------------------------------------------------------------------------------------------------------------------------------------------------------------------------------------------------------------------------|----------------------|-----------------------------------------------------------------|
|                                       | high_prec_timin g low_prec_freq low_prec_dur low_prec_timin g | season during which most high precipitation days (≥ 5 times mean daily precipitation) occur. If two seasons register the same number of events, a value of NaN is given.  frequency of dry days (< 1mm day <sup>-1</sup> )  average duration of dry periods (number of consecutive days < 1mm day <sup>-1</sup> )  season during which most dry days (< 1mm day <sup>-1</sup> ) occur. If two seasons register | days yr-1 days       |                                                                 |
|                                       | q_mean                                                        | the same number of events, a value of NaN is given.  mean daily discharge                                                                                                                                                                                                                                                                                                                                      | mm day <sup>-1</sup> | Catchment                                                       |
| Hydrologic Signatures (computed for   | runoff_ratio                                                  | runoff ratio, calculated as the ratio of<br>mean daily discharge to mean daily<br>precipitation                                                                                                                                                                                                                                                                                                                | -                    | timeseries of<br>streamflow and<br>precipitation                |
| 1st Oct 1970<br>to 30th Sept<br>2022) | stream_elas                                                   | streamflow precipitation elasticity (sensitivity of streamflow to changes in precipitation at the annual timescale, using the mean daily discharge as reference)                                                                                                                                                                                                                                               | -                    | described in Sections 3.1.2 and 3.1.1 respectively, and Table 2 |

|            | -1 6.1         | -1                                                        |                       |                   |
|------------|----------------|-----------------------------------------------------------|-----------------------|-------------------|
|            | slope_fdc      | slope of the flow duration curve (between                 | -                     |                   |
|            |                | the log-transformed 33 <sup>rd</sup> and 66 <sup>th</sup> |                       |                   |
|            |                | streamflow percentiles)                                   |                       |                   |
|            | baseflow_index | baseflow index (ratio of mean daily                       | -                     |                   |
|            |                | baseflow to daily discharge, hydrograph                   |                       |                   |
|            |                | separation performed using the Ladson et                  |                       |                   |
|            |                | al., 2013 digital filter)                                 |                       |                   |
|            | baseflow_index | baseflow index (ratio of mean daily                       | -                     |                   |
|            | _ceh           | baseflow to daily discharge, hydrograph                   |                       |                   |
|            |                | separation performed using the Gustard et                 |                       |                   |
|            |                | al., 1992 method described in Appendix                    |                       |                   |
|            |                | A)                                                        |                       |                   |
|            | hfd_mean       | mean half-flow date (date on which the                    | days                  |                   |
|            |                | cumulative discharge since 1 October                      | since 1st             |                   |
|            |                | reaches half of the annual discharge)                     | October               |                   |
|            | Q5             | 5% flow quantile (low flow)                               | mm day-1              |                   |
|            | Q95            | 95% flow quantile (high flow)                             | mm day-1              |                   |
|            | high_q_freq    | frequency of high-flow days (> 9 times                    | days yr <sup>-1</sup> |                   |
|            |                | the median daily flow)                                    |                       |                   |
|            | high_q_dur     | average duration of high flow events                      | days                  |                   |
|            |                | (number of consecutive days >9 times the                  |                       |                   |
|            |                | median daily flow)                                        |                       |                   |
|            | low_q_freq     | frequency of low flow days (< 0.2 times                   | days yr <sup>-1</sup> |                   |
|            |                | the mean daily flow)                                      |                       |                   |
|            | low_q_dur      | average duration of low flow events                       | days                  |                   |
|            |                | (number of consecutive days 

|                  | T               | -                                           | T                  | 1                  |
|------------------|-----------------|---------------------------------------------|--------------------|--------------------|
|                  | grass_perc_YY   | percentage cover of grass and pasture for   | %                  | 2018, 2019,        |
|                  | YY              | that year (YYYY)                            |                    | 2020, 2021,        |
|                  | shrub_perc_YY   | percentage cover of medium scale            | %                  | 2022 (Marston      |
|                  | YY              | vegetation (shrubs) for that year (YYYY)    |                    | et al., 2022,      |
|                  | crop_perc_YYY   | percentage cover of crops for that year     | %                  | 2024; Morton et    |
|                  | Y               | (YYYY)                                      |                    | al., 2020a, b, c,  |
|                  | urban_perc_YY   | percentage cover of suburban and urban      | %                  | 2021; Rowland      |
|                  | YY              | for that year (YYYY)                        |                    | et al., 2017,      |
|                  | inwater_perc_Y  | percentage cover of inland water for that   | %                  | 2020)              |
|                  | YYY             | year (YYYY)                                 |                    |                    |
|                  | bares_perc_YY   | percentage cover of bare soil and rocks for | %                  | 1                  |
|                  | YY              | that year (YYYY)                            |                    |                    |
|                  | sand_perc       | percentage sand                             | %                  |                    |
| Soil             | silt_perc       | percentage silt                             | %                  | 1                  |
| Attributes       | clay_perc       | percentage clay                             | %                  |                    |
| Each soil        | organic_perc    | percentage organic content                  | %                  | 1                  |
| attribute is     | bulkdens        | bulk density                                | g cm <sup>-3</sup> | European Soil      |
| accompanied      | tawc            | total available water content               | mm                 | Database           |
| by the           | porosity_cosby  | volumetric porosity (saturated water        | -                  | Derived Data       |
| percentage       |                 | content estimated using a pedotransfer      |                    | product            |
| missing and      |                 | function based on sand and clay fractions)  |                    | (Hiederer,         |
| the 5th, 50th    | porosity_hypres | volumetric porosity (saturated water        | -                  | 2013a, b), and     |
| and 95th         |                 | content estimated using a pedotransfer      |                    | the modelled       |
| percentile       |                 | function based on silt, clay and organic    |                    | depth to           |
| (apart from      |                 | fractions, bulk density and topsoil)        |                    | bedrock global     |
| percentage       | conductivity_co | saturated hydraulic conductivity            | cm h <sup>-1</sup> | product            |
| sand, silt, clay | sby             | (estimated using a pedotransfer function    |                    | (Pelletier et al., |
| and organic      |                 | based on sand and clay fractions)           |                    | 2016)              |
| content) of      | conductivity_hy | saturated hydraulic conductivity            | cm h <sup>-1</sup> | 2010)              |
| that attribute   | pres            | (estimated using a pedotransfer function    |                    |                    |
| across the       |                 | based on silt, clay and organic fractions,  |                    |                    |
| catchment        |                 | bulk density and topsoil)                   |                    |                    |
|                  | root_depth      | depth available for roots                   | m                  | 1                  |
| L                | 1               | I .                                         | l                  | I                  |

|              | soil_depth_pelle | depth to bedrock (maximum 50m)                   | m |                 |
|--------------|------------------|--------------------------------------------------|---|-----------------|
|              | tier             |                                                  |   |                 |
|              | inter_high_perc  | significant intergranular flow – high            | % |                 |
|              |                  | productivity                                     |   |                 |
|              | inter_mod_perc   | significant intergranular flow – moderate        | % |                 |
|              |                  | productivity                                     |   | British         |
|              | inter_low_perc   | significant intergranular flow - low             | % | Geological      |
|              |                  | productivity                                     |   | Survey          |
|              | frac_high_perc   | flow through fractures – high productivity       | % | hydrogeology    |
| Hydrogeolog  | frac_mod_perc    | flow through fractures – moderate                | % | map (BGS        |
| y Attributes |                  | productivity                                     |   | hydrogeology    |
|              | frac_low_perc    | flow through fractures – low productivity        | % | 625k, 2019) and |
|              | no_gw_perc       | rocks with essentially no groundwater            | % | superficial     |
|              | low_nsig_perc    | generally low productivity (intergranular        | % | deposits map    |
|              |                  | flow) but some not significant aquifer           |   |                 |
|              | nsig_low_perc    | generally not significant aquifer but some       | % |                 |
|              |                  | low productivity (intergranular flow)            |   |                 |
|              | station_type     | gauging station type denoted by the              | - |                 |
|              |                  | following abbreviations ( <b>B</b> Broad-crested |   |                 |
|              |                  | weir; C Crump profile single-crest weir;         |   |                 |
|              |                  | CB Compound broad-crested weir; CC               |   |                 |
|              |                  | Compound Crump weir; EM                          |   |                 |
|              |                  | Electromagnetic gauging station; EW              |   |                 |
|              |                  | Essex weir; FL Flume; FV Flat V                  |   | UK National     |
| Hydrometry   |                  | triangular profile weir; IV Index                |   | River Flow      |
| liyarometry  |                  | Velocity; MIS Miscellaneous; TP                  |   | Archive using   |
|              |                  | Rectangular thin-plate weir; US                  |   | the NRFA API*   |
|              |                  | Ultrasonic gauging station; VA Velocity-         |   |                 |
|              |                  | area gauging station; VN Triangular (V           |   |                 |
|              |                  | notch) thin-plate weir); XC Cross                |   |                 |
|              |                  | Correlation. Two abbreviations may be            |   |                 |
|              |                  | applied to each station relating to the          |   |                 |
|              |                  | measurement of low or high flows.                |   |                 |
|              |                  |                                                  |   |                 |

| daily flow peri   | first date that daily flow time series         | _                              |                            |
|-------------------|------------------------------------------------|--------------------------------|----------------------------|
| ·                 | •                                              |                                |                            |
| od_start          | provided in CAMELS-GB v2 is available          |                                | Catchment                  |
|                   | for this gauging station                       |                                | timeseries of              |
| daily_flow_peri   | end date that daily flow time series           | -                              | daily                      |
| od_end            | provided in CAMELS-GB v2 are                   |                                | streamflow                 |
|                   | available for this gauging station             |                                | described in               |
| daily_flow_perc   | percentage of days with flow time series       | %                              | Section 3.1.2              |
| _complete         | available from $1^{st}$ October $1970-31^{st}$ |                                | Section 5.1.2              |
|                   | September 2022                                 |                                |                            |
| hourly_flow_pe    | first date that hourly flow time series        | -                              |                            |
| riod_start        | provided in CAMELS-GB v2 is available          |                                | G + 1                      |
|                   | for this gauging station                       |                                | Catchment timeseries of    |
| hourly_flow_pe    | end date that hourly flow time series          | -                              |                            |
| riod_end          | provided in CAMELS-GB v2 are                   |                                | hourly                     |
|                   | available for this gauging station             |                                | streamflow<br>described in |
| hourly_flow_pe    | percentage of hours with flow time series      | %                              | Section 3.2.2              |
| rc_complete       | available from 1st October 1990 09:00:00       |                                | Section 3.2.2              |
|                   | - 1st October 2022 08:00:00                    |                                |                            |
| bankfull_flow     | flow at which the river begins to overlap      | $m^3 s^{-1}$                   |                            |
|                   | the banks at a gauging                         |                                |                            |
|                   | station (obtained from stage-discharge         |                                | UK National                |
|                   | relationships so may be derived by             |                                | River Flow                 |
|                   | extrapolation)                                 |                                | Archive using              |
| structurefull_flo | flow at which the river begins to the          | m <sup>3</sup> s <sup>-1</sup> | the NRFA                   |
| w                 | wingwalls of a structure at a gauging          |                                | API*,                      |
|                   | station (obtained from stage-discharge         |                                | catchment                  |
|                   | relationships so may be derived by             |                                | timeseries of              |
|                   | extrapolation)                                 |                                | streamflow                 |
| max_gauging_fl    | date and time when the maximum                 | -                              | described in               |
| ow_date           | gauging flow was taken                         |                                | Section 3.1.2              |
| max_gauging_fl    | the maximum gauging flow - the highest         | m <sup>3</sup> s <sup>-1</sup> | and 3.2.2                  |
| ow                | manual measurement of flow taken at a          |                                |                            |
|                   | gauging station                                |                                |                            |
|                   |                                                |                                |                            |

| <b>-</b> | T               | <u></u>                                     |   | •            |
|----------|-----------------|---------------------------------------------|---|--------------|
|          | max_daily_flow  | the maximum daily flow recorded in the      | % |              |
|          |                 | daily flow time series times provided in    |   |              |
|          |                 | CAMELS-GB v2                                |   |              |
|          | daily_flow_extr | the percentage of time (excluding NaNs)     | % |              |
|          | ap_dur          | that the daily flow timeseries is higher    |   |              |
|          |                 | than the maximum gauging flow               |   |              |
|          | max_hourly_flo  | the maximum hourly flow recorded in the     | % |              |
|          | w               | hourly flow time series times provided in   |   |              |
|          |                 | CAMELS-GB v2                                |   |              |
|          | hourly_flow_ext | the percentage of time (excluding NaNs)     | % |              |
|          | rap_dur         | that the hourly flow timeseries is higher   |   |              |
|          |                 | than the maximum gauging flow               |   |              |
|          | qXX_uncert_up   | upper bound of the discharge uncertainty    | % |              |
|          | per             | interval for the XX percentile flow given   |   |              |
|          |                 | as a percentage of the XX percentile flow   |   |              |
|          |                 | – estimates for XX values of 5, 25, 50, 75, |   |              |
|          |                 | 95, 99 are provided                         |   |              |
|          | qXX_uncert_lo   | lower bound of the discharge uncertainty    | % |              |
|          | wer             | interval for the XX percentile flow given   |   |              |
|          |                 | as a percentage of the XX percentile flow   |   |              |
|          |                 | – estimates for XX values of 5, 25, 50, 75, |   |              |
|          |                 | 95, 99 are provided                         |   | Derived from |
|          | quncert_meta    | metadata describing the reasons why         | - | Coxon et al  |
|          |                 | discharge uncertainty estimates are (not)   |   | (2015)       |
|          |                 | provided; Calculated discharge              |   |              |
|          |                 | uncertainties; No stage-discharge           |   |              |
|          |                 | measurements available; Less than 20        |   |              |
|          |                 | stage-discharge measurements                |   |              |
|          |                 | available for most recent rating;           |   |              |
|          |                 | Discharge uncertainty estimates not         |   |              |
|          |                 | provided as the estimated uncertainty       |   |              |
|          |                 | bounds were deemed to not accurately        |   |              |
|          |                 | reflect the discharge uncertainty or        |   |              |
| L        | 1               |                                             |   |              |

|            |                  | hagaing there was no somethle relationalist | 1        |                  |
|------------|------------------|---------------------------------------------|----------|------------------|
|            |                  | because there was no sensible relationship  |          |                  |
|            |                  | between stage and discharge.                |          |                  |
|            | benchmark_catc   | benchmark catchment (Y indicates the        | Y/N      | UK National      |
|            | h                | catchment is part of the UK Benchmark       |          | River Flow       |
|            |                  | Network, while N indicates that it is not)  |          | Archive;         |
|            |                  |                                             |          | Harrigan et al., |
|            |                  |                                             |          | (2018)           |
|            | surfacewater_ab  | mean surface water abstraction              | mm day-1 |                  |
|            | s                |                                             |          |                  |
|            | groundwater_ab   | mean groundwater abstraction                | mm day-1 |                  |
|            | S                |                                             |          |                  |
|            | discharges       | mean discharges (daily discharges into      | mm day-1 |                  |
|            |                  | water courses from water companies and      |          | 1 km × 1 km      |
|            |                  | other discharge permit holders reported to  |          | abstractions for |
|            |                  | the Environment Agency)                     |          | multiple         |
| Human      | abs_agriculture_ | percentage of total (groundwater and        | %        | purposes (csv    |
| Influence  | perc             | surface water) abstractions in catchment    |          | file) and 1 km × |
| Attributes |                  | for agriculture                             |          | 1 km             |
|            | abs_amenities_p  | percentage of total (groundwater and        | %        | discharges for   |
|            | erc              | surface water) abstractions in catchment    |          | multiple         |
|            |                  | for amenities                               |          | purposes         |
|            | abs_energy_per   | percentage of total (groundwater and        | %        | (RACT netcdf     |
|            | c                | surface water) abstractions in catchment    |          | file)            |
|            |                  | for energy production                       |          | (Rameshwaran     |
|            | abs_environmen   | percentage of total (groundwater and        | %        | et al., 2025)    |
|            | tal_perc         | surface water) abstractions in catchment    |          | et an, 2023)     |
|            |                  | for environmental purposes                  |          |                  |
|            | abs_industry_pe  | percentage of total (groundwater and        | %        |                  |
|            | rc               | surface water) abstractions in catchment    |          |                  |
|            |                  | for industrial, commercial and public       |          |                  |
|            |                  | services                                    |          |                  |
|            | J                |                                             |          |                  |

| Г.               |                                             |    |                            |
|------------------|---------------------------------------------|----|----------------------------|
| abs_watersuppl   | percentage of total (groundwater and        | %  |                            |
| y_perc           | surface water) abstractions in catchment    |    |                            |
|                  | for water supply                            |    |                            |
| abs_highloss_pe  | percentage of total (groundwater and        | %  |                            |
| rc               | surface water) abstractions in catchment    |    |                            |
|                  | that is classified as 'high loss'           |    |                            |
| abs_mediumloss   | percentage of total (groundwater and        | %  |                            |
| _perc            | surface water) abstractions in catchment    |    |                            |
|                  | that is classified as 'medium loss'         |    |                            |
| abs_lowloss_per  | percentage of total (groundwater and        | %  |                            |
| c                | surface water) abstractions in catchment    |    |                            |
|                  | that is classified as 'low loss'            |    |                            |
| abs_verylowloss  | percentage of total (groundwater and        | %  |                            |
| _perc            | surface water) abstractions in catchment    |    |                            |
|                  | that is classified as 'very low loss'       |    |                            |
| num_reservoir    | number of reservoirs in the catchment       | -  | UK Reservoir               |
| reservoir_cap    | total storage capacity of reservoirs in the | ML | Inventory                  |
|                  | catchment in megalitres                     |    | (Durant and                |
| reservoir_contri | percentage of the overall catchment         | %  | Counsell, 2018),           |
| buting_area      | surface area that is drained through        |    | SEPA's publicly            |
|                  | reservoirs                                  |    | available                  |
| reservoir_norma  | ratio of the capacity of a reservoir to the | -  | controlled                 |
| lised_upstream_  | average volume of precipitation received    |    | reservoirs                 |
| capacity         | by the catchment in a year                  |    | register                   |
| reservoir_he     | percentage of total reservoir storage in    | %  | (http://map.sepa           |
|                  | catchment used for hydroelectricty          |    | .org.uk/reservoir          |
| reservoir_nav    | percentage of total reservoir storage in    | %  | sfloodmap/Map.             |
|                  | catchment used for navigation               |    | htm, last access:          |
| reservoir_drain  | percentage of total reservoir storage in    | %  | 2 <sup>nd</sup> September, |
|                  | catchment used for drainage                 |    | 2025) and                  |
| reservoir_wr     | percentage of total reservoir storage in    | %  | Salwey et al.,             |
|                  | catchment used for water resources          |    | (2023)                     |
| <br><u>I</u>     |                                             | l  |                            |

|            |                  | 01                                          | 0./  |                  |
|------------|------------------|---------------------------------------------|------|------------------|
|            | reservoir_fs     | percentage of total reservoir storage in    | %    |                  |
|            |                  | catchment used for flood storage            |      |                  |
|            | reservoir_env    | percentage of total reservoir storage in    | %    |                  |
|            |                  | catchment used for environmental            |      |                  |
|            | reservoir_nouse  | percentage of total reservoir storage in    | %    |                  |
|            | data             | catchment where no use data were            |      |                  |
|            |                  | available                                   |      |                  |
|            | reservoir_year_f | year the first reservoir in the catchment   | -    |                  |
|            | irst             | was built                                   |      |                  |
|            | reservoir_year_l | year the last reservoir in the catchment    | -    |                  |
|            | ast              | was built                                   |      |                  |
|            | gw_well_id       | groundwater well identifier (corresponds    | -    |                  |
|            |                  | to the ID provided by the British           |      |                  |
|            |                  | Geological Survey)                          |      |                  |
|            | gw_well_name     | groundwater well name                       | -    |                  |
|            | gw_well_eastin   | groundwater well easting                    | -    |                  |
|            | g                |                                             |      | UK               |
|            | gw_well_northi   | groundwater well northing                   | -    | Hydrometric      |
|            | ng               |                                             |      | Register (Marsh  |
|            | gw_well_datum    | the altitude of the point from which        | mAOD | and Hannaford,   |
| Groundwate |                  | measurements are taken at                   |      | 2008)            |
| r Wells    |                  | a particular site                           |      |                  |
| r wens     | gw_well_depth    | depth of the groundwater well               | m    |                  |
|            | aquifer          | aquifer to which the water level variations | -    |                  |
|            |                  | in the wells are                            |      |                  |
|            |                  | attributed                                  |      |                  |
|            | daily_gwlevel_p  | first date that daily groundwater level     | -    |                  |
|            | eriod_start      | series provided in CAMELS-GB v2 is          |      | Groundwater      |
|            |                  | available for this groundwater well         |      | level timeseries |
|            | daily_gwlevel_p  | end date that daily groundwater level       | -    | described in     |
|            | eriod_end        | series provided in CAMELS-GB v2 is          |      | Section 4 4      |
|            |                  | available for this groundwater well         |      |                  |
|            | 1                |                                             | l    |                  |

| daily_gwlevel_p | percentage of days with groundwater       | - |  |
|-----------------|-------------------------------------------|---|--|
| erc_complete    | level data                                |   |  |
| monthly_gwlev   | first date that monthly groundwater level | - |  |
| el_period_start | series provided in CAMELS-GB v2 is        |   |  |
|                 | available for this groundwater well       |   |  |
| monthly_gwlev   | end date that monthly groundwater level   | - |  |
| el_period_end   | series provided in CAMELS-GB v2 is        |   |  |
|                 | available for this groundwater well       |   |  |
| monthly_gwlev   | percentage of months with groundwater     | - |  |
| el_perc_complet | level data                                |   |  |
| e               |                                           |   |  |