# Peer review of "CAMELS-GB v2: hydrometeorological time series and landscape attributes for 671 catchments in Great Britain"

_Earth System Science Data, 2025_

## Referee Comment (RC3)

**Review of manuscript essd-2025-608**

**Comments on the manuscript**

1. Quality control for daily and hourly streamflow:
   The manuscript states that both daily and sub-daily streamflow data are quality controlled by the source institutions prior to release, and that the hourly data originate from the quality-controlled UK-Flow15 dataset. At the same time, detailed quality control flags are only provided for the hourly data, while no equivalent flags or diagnostics are supplied for the daily streamflow. It would be helpful to clarify the rationale for this asymmetry.

   I suggest clarifying how users should interpret the statement that no data were removed or modified by the quality control process for the hourly data, given that the data have already undergone quality control within UK-Flow15. Is the term "quality control" used differently for the daily and hourly products? For example screening or filtering at the daily scale versus the provision of diagnostic flags without filtering at the hourly scale.

   In the manuscript, users are strongly encouraged to use the diagnostic flags for the hourly data to decide how to treat potentially problematic observations. While this flexibility is valuable, it may present a barrier for some users. I therefore suggest considering the provision of an additional, ready-to-use pre-filtered hourly streamflow product based on a simple and clearly documented flag selection. This would improve accessibility for users who prefer a conservative default dataset, while still retaining the full flagged dataset for more advanced or customised analyses.

2. Daily mean temperature definition (Line 134):
   The description of how daily mean temperature is calculated from HadUK-Grid data is potentially confusing. A brief explanation of the rationale for this convention, related to the timing of minimum temperature observations and the diurnal cycle, would help readers understand why this approach is used instead of averaging same-day maxima and minima.

3. Interpretation of daily PET from Hydro-PE (Line 151):
   The manuscript notes that several meteorological variables in the Hydro-PE HadUK-Grid dataset are temporally downscaled from monthly to daily resolution using smooth interpolation. While this approach is understandable for achieving long temporal coverage, it implies that daily PET and PETI values do not represent true day-to-day variability in all controlling variables. I suggest explicitly highlighting this

limitation and clarifying that the Hydro-PE daily PET is most appropriate for seasonal to long-term analyses rather than event-scale or short-term applications.

4. Groundwater well coverage:
The inclusion of groundwater level time series for 55 boreholes is a valuable new component of CAMELS-GB v2. At present, wells are restricted to those located within CAMELS-GB catchments. I suggest considering the inclusion of additional groundwater wells, if available, even if they do not fall strictly within CAMELS-GB catchment boundaries. Groundwater level time series can be highly valuable for hydrological studies even when not directly associated with a specific CAMELS-GB catchment.

5. Interpretation and flagging of land cover change data:
In Section 5.4, the manuscript notes that most catchments show a decrease in urban land cover from 2021 to 2022 that is unlikely to be reflected in the real world. If this artefact is known a priori, perhaps removing these data for the year 2022 would make sense. If not, it would at least be helpful to clarify how users should interpret these data. Options could be either explicitly flagging the affected years or transitions in the dataset or metadata, or providing clearer guidance in the manuscript and documentation on how these land cover time series should and should not be used for change detection analyses.

6. File naming conventions:
The manuscript does not explicitly describe the file naming convention for the time series files. A short note explaining the format would improve the ease of use, for example:

- camels_gb_v2_hydromet_daily_timeseries_{gauge_id}_{start_date}-{end_date}.csv

- camels_gb_v2_groundwater_daily_timeseries_{well_id}_{start_date}-{end_date}.csv

**Minor technical corrections in the manuscript**

1. Table 5, hydrometry attributes:
The description of structurefull_flow reads "flow at which the river begins to the wingwalls of a structure", which is missing a verb. This likely should read "flow at which the river begins to overtop / reach the wingwalls of a structure".

2. Line 37: Remove the extra "and" in "Global Streamflow  Indices and Metadata Archive".

3. Line 45: "finable" should be "findable".

4. Line 258: "Figure 4" is in bold, which is inconsistent with formatting elsewhere in the manuscript.

5. Benchmark catchments (Line 387):
   The sentence
   "All CAMELS-GB catchments are identified as either being part of this network or not to provide users with an indication of 'near-natural' catchments and suitable for studies where human impacts need to be minimal."
   is confusing. I suggest rephrasing to something like:
   "All CAMELS-GB catchments are flagged according to whether they belong to the UK Benchmark Network, providing users with an indication of which catchments are relatively near-natural and therefore more suitable for studies requiring minimal human impact."

6. Undefined abbreviation:
   The abbreviation "CEH" is used throughout the manuscript but is never defined. Please spell it out at first use as UK Centre for Ecology and Hydrology.

7. Equation numbering: Please consider numbering equations (in Section 5.8.3.)

8. Baseflow index description in Table 5:
   The description of baseflow_index_ceh refers to "the Gustard et al. (1992) method described in Appendix A". This appears to have been carried over from the CAMELS-GB v1 paper, as there is no Appendix A in the current manuscript.

**Comments regarding the dataset**

1. Data access and bulk download
   For ease of use, consider providing an option for bulk download that does not require users to create an account on the Environmental Information Platform or use wget, if this is feasible within platform constraints.

2. Hourly resampling and timestamp conventions:
   The manuscript states that 15-minute streamflow data were aggregated to hourly values using a next-hour resampling convention. It is unclear which timestamp convention is used for hourly precipitation. I suggest explicitly documenting the timestamp convention used for all hourly variables and ensuring consistency between streamflow and precipitation.

   More generally, adopting start-of-hour timestamps for all hourly variables would be consistent with the definition of the daily data, where values for a given date represent the mean or sum over that calendar day rather than a shifted window.

While timestamp conventions differ between regions and institutions, clear documentation and internal consistency are essential for correct interpretation and use of the data.

3. Units in Table 5:
   In Table 5, max_daily_flow and max_hourly_flow are listed with units of percent. These appear to be absolute flow values and should likely have units of mm day$^{-1}$, mm hour$^{-1}$, or m$^3$ s$^{-1}$. Please clarify and correct if needed.

4. Consistency of flow units:
   Flow-related attributes in Table 5 use a mix of mm day$^{-1}$ and m$^3$ s$^{-1}$. While this reflects common practice, it may be worth considering whether greater standardisation would improve usability.

5. Runoff ratios greater than one:
   Seven catchments have runoff ratios exceeding 1.0, notably:

- gauge_id 26006 with a runoff ratio of 3.01

- gauge_id 27038 with a runoff ratio of 2.74
  I suggest explicitly identifying these catchments in the manuscript and briefly discussing potential explanations, to alert users.

6. Missing value conventions:
   For the attribute high_prec_timing, Table 5 states that NaN is used when two seasons register the same number of events. However, the file camels_gb_v2_climatic_attributes.csv uses "NA" instead of "NaN". I suggest changing this to NaN for consistency.

7. Attributes with high fractions of missing values:
   Some hydrometry attributes have more than 50 percent missing values, notably bankfull_flow (53 percent missing) and structurefull_flow (65 percent missing). A brief explanation of the reasons for this level of missingness would help users assess the reliability and appropriate use of these attributes.

8. Shapefiles for monitoring locations:
   The dataset includes catchment boundary shapefiles but does not appear to include shapefiles of streamflow gauge locations or groundwater well locations. Providing point shapefiles for both river gauging stations and groundwater wells would be beneficial. While the relevant coordinates are available in tabular form, distributing these locations as shapefiles would be consistent with the provision of catchment boundaries and common practice in large-sample hydrology datasets.

Including key attributes directly in the shapefiles (also the catchment boundaries shapefile) would further enhance usability.